# Country of birth as a potential determinant of inadequate antenatal care use among women giving birth in Brussels. A cross-sectional study

Claudia Schönborn[1,2]*, Katia Castetbon[3], Myriam De Spiegelaere[1]

1 Research Centre in Social Approaches to Health, School of Public Health, Université libre de Bruxelles, Brussels, Belgium, 2 Research Centre in Health Systems and Policies, School of Public Health, Université libre de Bruxelles, Brussels, Belgium, 3 Research Centre in Epidemiology, Biostatistics and Clinical Research, School of Public Health, Université libre de Bruxelles, Brussels, Belgium

* claudia.schoenborn@ulb.be

## Abstract

### Background

One of the mechanisms explaining perinatal health inequalities could be inadequate antenatal care among some immigrant groups. Few European studies compared antenatal care use between different groups of immigrants taking into account individual characteristics. This research investigates the associations of three birth regions with the use of antenatal care, by also considering socioeconomic and migration-related determinants.

### Methods

We included 879 mothers born in Belgium, North Africa, and Sub-Saharan Africa, and interviewed them in four Brussels hospitals after they gave birth, using an adapted version of the Migrant-Friendly Maternity Care Questionnaire. We additionally collected clinical data from hospital records. We carried out descriptive analyses and ran univariate and multivariable logistic regression models to estimate the associations of socioeconomic and migration characteristics with a) late start of antenatal care and b) less than minimum recommended number of consultations.

### Results

The vast majority of women in this study had adequate care in terms of timing (93.9%), frequency of consultations (82.2%), and self-reported access (95.9%). Region of birth was an independent risk factor for late initiation of care, but not for infrequent consultations. Women born in Sub-Saharan Africa were more prone to accessing care late (OR 3.3, 95%CI 1.5–7.7), but were not more at risk of infrequent consultations. Women born in North Africa, had similar adequacy of care compared to the Belgium-born population. The three groups also differed in terms of socioeconomic profiles and socioeconomic predictors of antenatal care

**Data Availability Statement:** We unfortunately cannot publicly share our dataset due to ethical restrictions. In fact, our dataset contains sensitive data, such as sex (the study is restricted to

women), age, country of birth, nationality, legal status, delivery hospital and delivery year, detailed clinical data concerning the mother and their child, detailed socioeconomic data such as health insurance status, social welfare status, professional activity etc. Although not directly identifying, in combination these data are likely to become identifying. The Research Ethics Committee has imposed this restriction. However, we are happy to make a portion of the database available upon reasonable request to interested and qualified researchers. Data requests may be sent to the principal investigator (CS) or to the Data Protection Unit: Data Protection Unit, Université libre de Bruxelles, Campus du Solbosch - CP 130 Avenue F.D. Roosevelt, 50, 1050 Bruxelles, e-mail : rgpd@ulb.be.

**Funding:** The funder (FNRS) provided support in the form of salary for authors [CS], but did not have any additional role in the study design, data collection and analysis, decision to publish, or preparation of the manuscript. The specific roles of the authors are articulated in the 'author contributions' section. Furthermore, the funder (ULB) provided support in the form of funding for data collection and materials, e.g. remuneration for interviewers, tablets, printing costs [MDS, CS], but did not have any additional role in the study design, data collection and analysis, decision to publish, or preparation of the manuscript. The specific roles of the authors are articulated in the 'author contributions' section.

**Competing interests:** The authors have declared that no competing interests exist.

use. Housing type, professional activity, and health insurance status were important predictors of both outcomes.

## Conclusions

This study showed that the region of birth was partly associated with adequacy of care, in terms of initiation, but not number of consultations. Further dimensions of adequacy of care (content, quality) should be studied in the future. Socioeconomic factors are also key determinants of antenatal care use.

## Background

Perinatal health inequalities in relation to migration have been widely reported across high-income countries [1], with many studies reporting increased risks among immigrants [1, 2]. It is becoming apparent, however, that the impact of migration on perinatal health largely depends on the specific immigrant groups, on the outcome studied, and on the geographic location [1, 3]. Populations which stand out as being particularly at risk are non-European migrants to Europe, refugees and asylum-seekers, and foreign-born blacks in the U.S. [4].

In Brussels, studies have shown that mothers with a nationality from the two largest non-European immigrant groups, namely North- and Sub-Saharan Africa, experienced around 80% higher perinatal mortality rates compared to Belgians [5, 6]. Women with a Sub-Saharan African nationality were also at higher risk of giving birth preterm, whereas women from North Africa were less at risk for this outcome, compared to Belgian mothers [5, 6].

Some studies found that perinatal health inequalities between immigrant and non-immigrant populations were no longer present when adjusting for socioeconomic determinants [7, 8]. Although this effect might suggest that the socioeconomic disadvantage of immigrants might fully account for the observed inequalities in health, other studies have concluded that additional mechanisms must be at play [9, 10].

Authors have suggested that inadequate antenatal care among certain immigrant groups could be one of the key mechanisms that ought to be explored [9]. Indeed, timely antenatal care is known to prevent pregnancy complications (through screening and treatment, health advice and psychological support), it can make pregnancy a more positive experience, and can improve maternal and perinatal outcomes [11, 12]. However, studies have shown that immigrants or ethnic minorities in high-income countries may experience increased access barriers, use services less, or be more exposed to receiving suboptimal care [2, 9, 13].

Furthermore, as is the case with some perinatal health outcomes such as prematurity, the approach of comparing native women with immigrants might hide crucial differences within the immigrant population. Few studies compared the antenatal care (ANC) use between different groups of immigrants, and most were mainly carried out in North America. Findings from these studies indicated that important variations did exist, either by country of birth or by legal status [2, 13–15]. Two studies carried out in the UK showed that health-seeking behaviours were determined by socio-cultural background and familiarity with the healthcare system. The prevalence of late initiation of care varied by ethnic groups, and was highest among women of South Asian, African Caribbean, or Eastern European ethnicity born abroad. The reasons for not accessing care differed between groups: e.g., Somali and Bengali were concerned ANC would be over-medicalised, whereas Eastern European women were worried it would not be medicalised enough [16]. A French prospective study including nearly 10 000 women found

that, compared to other regions of birth, women born in Sub-Saharan Africa were more likely to have inadequate antenatal care [14].

Furthermore, since socioeconomic deprivation has been associated with inadequate antenatal care on one hand [17], and given its strong link to immigration on the other hand, it seems essential that studies take into account the socioeconomic characteristics of immigrants. Such an approach enables to better understand the relationship of country of birth with the use of antenatal care, and to estimate the potential intermediate role of other determinants in this relationship.

This research investigates the associations of three birth regions (Belgium, North Africa, and Sub-Saharan Africa) with the use of antenatal care, by also considering demographic, socioeconomic and migration-related determinants.

## Methods

### Population and tools

This observational cross-sectional study included 932 women interviewed in hospital within two weeks of having given birth. The detailed methodology has been published elsewhere [18].

The sampling focused on women with a Belgian nationality and on those with a nationality from the two largest non-European nationality groups in terms of births in Brussels, namely North and Sub-Saharan Africa. Between January 2019 and February 2020, we recruited women in four hospitals in Brussels. We selected the hospitals with the highest proportions of women with an African nationality, two with more socially disadvantaged patients and two with more affluent ones, thus aiming to include a range of socioeconomic situations.

We invited all mothers with a current nationality from the above three groups (based on UN country classification [19], S1 Table), who were 16 years old or above, and admitted on the maternity or neonatal ward, to participate in the survey. We included all women, regardless of health insurance, legal status, or literacy. Women speaking French, Dutch, Arabic, Riff, Peul, English or Spanish were eligible. Women who were considered by their midwife to be too unwell to be approached were not asked to participate.

Data was collected via an adapted version of the Migrant-Friendly Maternity Care Questionnaire [20] (S1 File), which was administered face-to-face by trained polylingual female interviewers. The questionnaire was translated into English (S2 File), and orally into Moroccan dialect. Translation into other languages was done *ad hoc*. Nearly all questionnaires were administered in French (86%) and Arabic (13%). Few questionnaires were administered in English (n = 4), or translated ad hoc into Peul (n = 2), Riff (n = 3), or Dutch (n = 2).

For participants who additionally consented to give access to their hospital records, we also collected clinical data, including complications during pregnancy, birth and postpartum, timing of first antenatal consultation and number of consultations.

We obtained ethical approval from the heads of Obstetrics and of the Ethics Committees of all four hospitals and from the Université libre de Bruxelles (CHU Erasme, Reference No P2017/055/B406201730877), which included approval of interviewing women from the age of 16 without parent or guardian consent. Oral and written information were given to all participants and written consent was received.

### Setting

In Brussels, three-quarters of births are to mothers with an immigrant background [21], and around 1.9% are to women who are not in the Belgian register (mostly undocumented) [22]. Different settings are available for the clinical follow-up of pregnancy, including hospitals, private practices, and community perinatal centres [23]. Mothers-to-be may freely choose

between these settings. Antenatal care is mostly provided by obstetricians, midwives or both; with only a tiny proportion followed-up by GPs. Ninety-eight percent of the population legally residing in Belgium has health insurance [24], which covers an important part of perinatal healthcare costs. Midwifery consultations are free of charge for the insured population, however obstetrical consultations are charged, and then partly reimbursed by health insurance. There are various measures aimed at facilitating vulnerable pregnant women's access to antenatal care. Free clinical follow-up is provided, mainly in hospital, via the ONE (Office for Birth and Childhood), which, in 2016, accounted for 55% of pregnancy follow-up in Brussels [23]. For women without health insurance, perinatal care can be financed by social services in the case of legal residents, by Urgent Medical Care in the case of undocumented immigrants, or by the Federal Agency for the reception of asylum seekers.

## Measurements

We collected data on several aspects of care received during pregnancy, labour and postpartum. We also collected detailed maternal demographic, socioeconomic, migration and obstetric characteristics, health behaviours and language competency.

Our two main outcome variables were the late initiation of antenatal care and not having received the minimum recommended number of consultations. We considered a first consultation as being late from 14 weeks gestation or after the first three pregnancy months. We used the self-reported timing of initiation of care, and missing values (n = 4) were replaced with the timing extracted from clinical records.

We asked mothers how many antenatal care visits they had had with a gynaecologist or a midwife, excluding the visits in which they only had an ultrasound scan without a consultation. The answer categories were: no care/<3 /3-6 /7-9 /10+. When women replied "don't know" (n = 5), we used the value extracted from clinical records. The Belgian Healthcare Knowledge Centre recommends that primipara have a minimum of 10 antenatal care visits and that multipara, a minimum of 7 [25]. In order to assess whether women received what is recommended, we combined the declared number of visits with their parity, and categorised women into having or not having received the minimum recommended number of consultations.

Region of birth was categorised into three: Belgium (n = 422), North Africa (n = 263), and Sub-Saharan Africa (n = 194), thus excluding those born outside these three regions (20 born in EU27 and 33 born elsewhere).

Three groups were defined for the level of education: women who had completed at the most three years of secondary school, those who finished secondary school, and those who had a university or higher education degree. The variable of single motherhood distinguished between women who reported being married or in a couple, and those who were single. To assess housing type, participants were asked what kind of accommodation they lived in. There were six answer options: private (as owners or tenants), social housing, living at family's house, living at friends' house, living in an immigration centre, being homeless. The latter 4 categories were grouped together as "not own accommodation". Professional activity related to the last trimester of pregnancy, and was classified as follows: 1) working, having a job but being on pregnancy leave, or student; 2) recipient of social welfare of last resort, on unemployment benefits, or work incapacity; 3) no professional activity or housewife. Participants were asked how much their net monthly household income was in euros (0–500; 500–1000; 1000–1500; 1500–2000; 2000–3000; 3000–4000; >4000). Based on the OECD modified scale [26], we computed an equivalised household income which factors in household size (attributing a weight to each cohabiting person, depending on age). The middle values of the income categories were used,

along with the responses to an item asking participants about the number of adults and of children they lived with.

Women having either public or private health insurance were categorised as having health insurance. We categorised as not having health insurance those women who accessed care through the Public Social Welfare Centre (a service to help people gain access to healthcare), "Urgent Medical Care" (access to healthcare for undocumented migrants), Fedasil (access for asylum seekers), or if they had to pay out of pocket.

We asked participants born abroad and those born in Belgium who did not have the Belgian nationality at birth what their immigration status was. We classified as "stable legal status" women with a current Belgian nationality, EU citizens, those with indeterminate residence permit, family reunion, and refugees. We considered "unstable" those with short stay visas, asylum seekers, and undocumented women.

Women were asked to self-rate their oral French and Dutch language competencies, the official languages in Belgium. For each language, we proposed the following answer options: fluent, good, with difficulty, not at all. We grouped the last two categories together, and combined the answers for the two languages by only considering the category for the language that was spoken best. For instance, if a woman spoke French fluently and Dutch with difficulty, she was classified into "fluent".

Continuity of care was assessed through the combination of answers to two questions: "Have you always or almost always been followed-up by the same gynaecologist or midwife?", and to those answering "no": "Did you feel that the different healthcare professionals ensured a continuity in your care by sharing information or working well together?" We classified continuity into three categories: 1) always/almost always seen by same healthcare professional 2) not seen by same healthcare professional but continuity in information 3) not seen by same healthcare professional and only some or no continuity in information.

We classified as having had medical complications during their pregnancy those women whose clinical records mentioned at least one of the following conditions: anaemia, hypertension, pre-eclampsia, premature labour, deep vein thrombosis, gestational diabetes, placenta praevia, placental abruption, urinary tract infection, severe back pain, premature rupture of membranes, depression, hyperemesis, congenital foetal anomaly, or other condition. In addition, we included in this category those women who declared having had one of these complications in the questionnaire item, unless they mentioned only a single minor complication (anaemia, back pain, UTI) which was not mentioned in the clinical notes (n = 55). When data from clinical records was missing (n = 82), we used women's responses instead.

We also extracted questionnaire answers for: age, parity, time of residence in Belgium, type of follow-up, wish to have had more consultations. Response categories for these items are shown in the results tables.

## Statistical analysis

We carried out descriptive analyses (frequencies) of demographic, socioeconomic, and migration-related indicators, for the whole sample and stratified by region of birth. We calculated Chi$^2$ or Fisher exact p-values, as appropriate, to determine whether distributions were different between birth regions. The same method was used for variables related to antenatal care.

We ran univariate logistic regressions to estimate the associations of demographic, socioeconomic and migration characteristics with a) late start of ANC and b) less than minimum recommended number of consultations (sometimes shortened to "infrequent consultations"). Given that the first outcome (late start) had a statistically significant association with region of birth, we explored this relationship further by carrying out multivariable logistic regression to

adjust for demographic, socioeconomic and migration characteristics. We didn't include variables which were strongly associated with another predictor (Cramer V value >0.25) and which were conceptually redundant, in order to keep variables which were as complementary as possible. Given that an estimate of household income was missing for fifty women in the sample, we carried out a sensitivity analysis by removing it from the model, and found that the results did not change significantly. Running multivariable models stratified by region of birth would have enabled us to assess whether the role of socioeconomic factors differed by birth region; however, it was not possible to run such models because of the insufficient number of subjects in each category.

We further analysed the second outcome, infrequent consultations, by adjusting for the start of antenatal care (first trimester versus later) and gestation at delivery (<37; 37–40; 41–42 weeks).

The Hosmer and Lemeshow test was used to check the suitability of the models. Analyses were processed through Stata, version 14.

## Results

### Participation rates & sample description

Our sample consisted of 932 women who fully answered the questionnaire. For 850 of these, additional clinical data was collected from hospital records. The overall participation rate was 86.4% (varying between 80.3 and 90.2% for the different nationality groups, and between 82.6 and 88.9% for the four hospitals). Our analyses concern the 879 women born in one of the three birth regions of interest.

Ninety percent of women born in North Africa were born in Morocco, 7% in Algeria, 2% in Tunisia and 1% in Lybia or Egypt. There were 20 Sub-Saharan African countries of birth, with the most represented being the Democratic Republic of Congo (31.4%), Guinea (19.6%), and Cameroon (16.0%), followed by Rwanda (5.2%), Senegal (4.7%), and Côte d'Ivoire (4.2%).

Women born in Belgium were younger, had lower parity, and experienced fewer medical complications during pregnancy than immigrant women (**Table 1**). Mothers born in North Africa were characterised by older age, lower education, high proportions without a professional activity, and lowest proportions in the wealthiest quartile. They were also particularly likely to have difficulties speaking the host-country languages. Mothers born in Sub-Saharan Africa had the highest proportions of grand multipara and single mothers. They were most at risk of not having their own accommodation, and poverty was prominent with almost half of women having a household income in the lowest quartile. Women from Sub-Saharan Africa were most prone to lacking health insurance, and to having an unstable legal status.

### Antenatal care

Around half of the women in the sample were followed-up mainly by an obstetrician, and 13.2% mainly by a midwife (**Table 2**). The vast majority had their first antenatal consultation during the first trimester of pregnancy (93.9%), and the proportion of late initiation was distinctively higher among women born in Sub-Saharan Africa (14.4%). Among late starters, most had their first visit no later than the fourth pregnancy month, with proportions very high for women born in North Africa. The three main reasons that women cited for consulting late were, in order of prevalence: 1) not knowing that they were pregnant (44%), 2) not feeling the need to consult (16%), 3) difficulty getting an appointment earlier (12%). The reasons given by women who did not consult at all during pregnancy (n = 5) were: 1) being scared to be reported (because undocumented) and not being aware of their rights, 2) having serious family problems overtaking the priority of consulting, and 3) being in denial of pregnancy.

**Table 1. Demographic, clinical, and socioeconomic characteristics of women (in %), overall and by region of birth, with X² p-values.**

| | | Total (n = 879) | Belgium (n = 422) | North Africa (n = 263) | Sub-Saharan Africa (n = 194) | p-value |
|---|---|---|---|---|---|---|
| **Age** | | | | | | <0.001 |
| | <25 years | 9.7 | 12.6 | 6.5 | 7.7 | |
| | 25–35 years | 64 | 65.9 | 55.9 | 70.6 | |
| | 36+ years | 26.4 | 21.6 | 37.6 | 21.7 | |
| **Parity** | | | | | | <0.001 |
| | 1 | 36.3 | 44.1 | 28.1 | 30.4 | |
| | 2–3 | 50.1 | 46.9 | 56.3 | 48.5 | |
| | 4+ | 13.7 | 9.0 | 15.6 | 21.1 | |
| **Pregnancy complications** | | | | | | 0.09 |
| | Yes | 56.2 | 52.4 | 59.3 | 60.3 | |
| **Education** | | | | | | <0.001 |
| | Lower secondary or less | 27.1 | 15.6 | 37.8 | 37.8 | |
| | Upper secondary | 36.2 | 35.3 | 39.7 | 33.2 | |
| | Higher education | 36.7 | 49.1 | 22.5 | 29.0 | |
| **Single motherhood** | | | | | | <0.001 |
| | Yes | 12.8 | 10.9 | 3.1 | 30.1 | |
| **Housing type** | | | | | | <0.001 |
| | Private housing | 83.7 | 88.9 | 82.7 | 73.6 | |
| | Social housing | 10.9 | 6.6 | 14.2 | 15.5 | |
| | Not own home | 5.5 | 4.5 | 3.1 | 10.9 | |
| **Professional activity** | | | | | | <0.001 |
| | Employed/student | 55 | 72.3 | 29.4 | 51.8 | |
| | Unemployed/social welfare/invalidity | 15.6 | 16.1 | 11.1 | 20.7 | |
| | None/housewife | 29.4 | 11.6 | 59.5 | 27.5 | |
| **Equivalised household income** | | | | | | <0.001 |
| | Q1 (lowest) | 29.7 | 14.7 | 41.2 | 49.2 | |
| | Q2 | 25.2 | 21.3 | 30.9 | 26.8 | |
| | Q3 | 26.5 | 32.1 | 23.2 | 17.9 | |
| | Q4 (highest) | 18.6 | 31.9 | 4.7 | 6.2 | |
| **Health insurance** | | | | | | <0.001 |
| | No | 5.5 | 0.5 | 5.3 | 16.6 | |
| **Time of residence** | | | | | | <0.001 |
| | < = 5 years | 19.2 | 0.2 | 35.7 | 38.1 | |
| | 6–15 years | 24.8 | 0.7 | 47.5 | 46.4 | |
| | 16+ years | 56 | 99.1 | 16.7 | 15.5 | |
| **Legal status** | | | | | | <0.001 |
| | unstable | 6.6 | 0.0 | 5.7 | 22.2 | |
| **Language** | | | | | | <0.001 |
| | Fluent | 73.8 | 99.1 | 39.2 | 66.0 | |
| | Good | 14.1 | 1.0 | 27.0 | 25.3 | |
| | Difficult/not at all | 12.1 | 0.0 | 33.8 | 8.8 | |

Overall, only 5.2% of women had less than seven antenatal consultations, and 17.8% had less than the recommended number of consultations with respect to their parity. The prevalence of "infrequent consultations" was much higher in primipara (37.9%), than in multipara (6.3%). In fact, primipara having had between 7 and 9 ANC visits constituted 71% of women with "infrequent consultations", multipara with <7 visits 22%, and primipara with less than 7

**Table 2. Antenatal care characteristics, overall and by region of birth, with X² p-values.**

| | Total (n = 879) | | Belgium (n = 422) | | North Africa (n = 263) | | Sub-Saharan Africa (n = 194) | | |
|---|---|---|---|---|---|---|---|---|---|
| Characteristic | n | % | n | % | n | % | n | % | p-value |
| **Type of follow-up (n = 874)** | | | | | | | | | 0.59 |
| Obstetrician | 485 | **52.4** | 226 | **53.8** | 133 | **50.6** | 96 | **50.5** | |
| Midwife | 122 | **13.2** | 58 | **13.8** | 37 | **14.1** | 21 | **11.1** | |
| Both | 319 | **34.5** | 136 | **32.4** | 93 | **35.4** | 73 | **38.4** | |
| **Timing of first consultation (n = 879)** | | | | | | | | | <0.001 |
| > 1st trimester | 54 | **6.1** | 14 | **3.3** | 12 | **4.6** | 28 | **14.4** | |
| **Timing of first consultation if > 1st trimester (n = 54)** | | | | | | | | | |
| 4th month | 34 | **63.0** | 7 | **50.0** | 11 | **91.7** | 16 | **57.1** | |
| 5th month | 8 | **14.8** | 4 | **28.6** | 0 | **0** | 4 | **14.3** | |
| 6th month or later | 7 | **13.0** | 1 | **7.1** | 1 | **8.3** | 5 | **17.9** | |
| no care | 5 | **9.3** | 2 | **14.3** | 0 | **0** | 3 | **10.7** | |
| **Number of antenatal consultations (n = 879)** | | | | | | | | | 0.09 |
| <7 | 46 | **5.2** | 17 | **4.0** | 12 | **4.6** | 17 | **8.8** | |
| 7–9 | 327 | **37.2** | 164 | **38.9** | 101 | **38.4** | 62 | **32.0** | |
| 10+ | 506 | **57.6** | 241 | **57.1** | 150 | **57.0** | 115 | **59.3** | |
| **Min. recommended ANC (n = 879)** | | | | | | | | | 0.27 |
| No | 156 | **17.8** | 84 | **19.9** | 42 | **16.0** | 30 | **15.5** | |
| **Difficulty to get an appointment (n = 874)** | | | | | | | | | 0.68 |
| Yes | 36 | **4.1** | 16 | **3.8** | 10 | **3.8** | 10 | **5.2** | |
| **Wanted more appointments (n = 874)** | | | | | | | | | 0.20 |
| Yes | 61 | **6.6** | 22 | **5.2** | 20 | **7.6** | 17 | **8.9** | |
| **Continuity of care (n = 879)** | | | | | | | | | 0.54 |
| Yes | 713 | **76.5** | 327 | **77.5** | 206 | **78.3** | 140 | **72.2** | |
| Medium | 154 | **16.5** | 65 | **15.4** | 41 | **15.6** | 39 | **20.1** | |
| No | 65 | **7.0** | 30 | **7.1** | 16 | **6.1** | 15 | **7.7** | |

visits only 7%. Few women said that they had experienced difficulties getting an appointment or that they would have liked to have more antenatal consultations. Seven percent of women said they had had no continuity of care.

**Predictors of late start of ANC.** Univariate analyses (Table 3) show that women born in Sub-Saharan Africa had almost five times higher odds of starting ANC late, compared to women born in Belgium, whereas no statistically significant difference was found for women born in North Africa.

Other characteristics were identified as important predictors for late start of ANC. Of note, mothers not having their own home had 12 times higher odds of starting ANC after the first trimester compared to women who lived in private housing; and women without health insurance had 8 times higher odds. Other significant predictors of delayed start of ANC were being in the lowest income quartile, in the lowest education category, single motherhood, and having an unstable immigration status. Moreover, women who were unemployed or without a professional activity were around 4 times more likely to start ANC late than women with a job or

**Table 3. Crude and adjusted odds ratios and p-values for predictors of late start of care.**

| Variable | n | % late start | ORc | p-value | ORa (n = 824) | p-value |
|---|---|---|---|---|---|---|
| **Region of birth** | | | | **<0.001** | | **0.001** |
| Belgium | 422 | 3.3 | 1 | | 1 | |
| North Africa | 263 | 4.6 | 1.39 (0.63–3.06) | | 0.75 (0.26–2.13) | |
| Sub-Saharan Africa | 194 | 14.4 | **4.92 (2.52–9.57)** | | **3.33 (1.45–7.68)** | |
| **Age** | | | | 0.63 | | 0.38 |
| <25 years | 85 | 4.7 | 0.79 (0.27–2.29) | | 0.37 (0.09–1.52) | |
| 25–35 years | 562 | 5.9 | 1 | | 1 | |
| 36+ years | 232 | 7.3 | 1.27 (0.69–2.32) | | 0.89 (0.41–1.94) | |
| **Parity** | | | | 0.28 | | 0.27 |
| 1 | 319 | 5 | 1 | | 1 | |
| 2–3 | 440 | 6.1 | 1.24 (0.66–2.34) | | 1.31 (0.56–3.06) | |
| 4+ | 120 | 9.2 | 1.91 (0.86–4.26) | | 2.31 (0.81–6.60) | |
| **Education** | | | | **<0.001** | | 0.28 |
| = < lower secondary | 238 | 12.6 | **5.66 (2.55–12.59)** | | 2.16 (0.75–6.23) | |
| Upper secondary | 317 | 5.1 | 2.09 (0.88–4.95) | | 1.35 (0.48–3.79) | |
| Higher education | 322 | 2.5 | 1 | | 1 | |
| **Single motherhood** | | | | **<0.001** | | 0.29 |
| No | 765 | 4.4 | 1 | | 1 | |
| Yes | 112 | 17.9 | **4.67 (2.58–8.46)** | | 1.64 (0.66–4.11) | |
| **Housing type** | | | | **<0.001** | | **<0.001** |
| Private housing | 732 | 4 | 1 | | 1 | |
| Social housing | 95 | 9.5 | **2.54 (1.16–5.54)** | | 1.66 (0.66–4.18) | |
| Not own home | 48 | 33.3 | **12.12 (5.99–24.55)** | | **10.12 (3.26–31.39)** | |
| **Professional activity** | | | | **<0.001** | | **0.03** |
| Employed /student | 482 | 2.9 | 1 | | 1 | |
| Unemployed/invalidity | 137 | 11 | **4.11 (1.93–8.75)** | | **2.93 (1.18–7.32)** | |
| None | 258 | 9.7 | **3.59 (1.83–7.03)** | | **2.82 (1.17–6.79)** | |
| **Equivalised income** | | | | **<0.001*** | | 0.61 |
| Q1 (lowest) | 245 | 10.2 | **5.72 (1.70–19.28)** | | 0.52 (0.10–2.62) | |
| Q2 | 208 | 5.8 | 3.08 (0.85–11.12) | | 0.83 (0.18–3.87) | |
| Q3 | 219 | 3.7 | 1.91 (0.50–7.31) | | 1.01 (0.23–4.42) | |
| Q4 (highest) | 154 | 2 | 1 | | 1 | |
| **Health insurance** | | | | **<0.001** | | 0.52 |
| Yes | 829 | 4.8 | 1 | | 1 | |
| No | 48 | 29.2 | **8.12 (4.04–16.34)** | | 1.42 (0.49–4.15) | |
| **Duration of residence** | | | | **<0.05** | | |
| <5 years | 169 | 10.1 | **2.51 (1.29–4.88)** | | | |
| 5–15 years | 218 | 7.3 | 1.78 (0.91–3.48) | | | |
| 16+ years | 492 | 4.3 | 1 | | | |
| **Legal status** | | | | **<0.001** | | |
| Stable | 821 | 5.2 | 1 | | | |
| Unstable | 58 | 19 | **4.23 (2.05–8.74)** | | | |
| **Language** | | | | 0.06* | | |
| Fluent | 649 | 5.4 | 1 | | | |
| Good | 124 | 6.5 | 1.21 (0.55–2.68) | | | |

(*Continued*)

**Table 3.** (Continued)

| Variable | n | % late start | ORc | p-value | OR$_a$ (n = 824) | p-value |
|---|---|---|---|---|---|---|
| With difficulty/not at all | 106 | 10.4 | 2.03 (1.00–4.14) | | | |

*Test for trend

ORa: adjusted for region of birth, age, parity, education, single motherhood, housing type, professional activity, household income, health insurance status.

students. Recently arrived immigrants were also more at risk of starting ANC after the first trimester, compared to women having resided in Belgium for at least 16 years.

In the multivariable model, country of birth maintained its statistically significant association with late start of care. The adjusted results showed that Sub-Saharan-born mothers had 3 times higher odds of initiating ANC late compared with non-migrants. Accommodation type and professional activity also remained strongly associated with the outcome, with women without their own home, unemployed and women without a professional activity having increased odds of initiating care late. Generally speaking, confidence intervals were large due to the small numbers of women in these categories.

**Table 4** shows the prevalence of late initiation of care according to sociodemographic, socioeconomic and migration characteristics, stratified by region of birth. Among women born in Belgium, the statistically significant indicators for late start were the level of education, single motherhood, housing type, and professional activity. Among mothers born in North Africa, only age, professional activity, and health insurance status were associated with late start. However, for women born in Sub-Saharan Africa, numerous socioeconomic determinants predicted the delayed start of care: the predictors were the same as those for Belgium-born women, with the addition of health insurance and household income.

**Predictors of "infrequent consultations".** Parity, housing type, and late initiation of ANC were the strongest predictors for having less than the recommended number of visits (**Table 5**). Despite having initiated care late, more than half (60%) of late starters born in North- and Sub-Saharan Africa had the recommended number of consultations. When adjusting for late initiation and gestation at delivery, women born in Sub-Saharan Africa were less likely to have insufficient consultations compared to women born in Belgium. The adjusted odds ratios show that primipara had 10 times higher odds of having less than the recommended number of consultations, compared to multipara. Mothers without their own accommodation had 5 times the odds compared to those in private housing. Other at-risk categories were mothers without health insurance, single mothers, and those under 25 years, those with unstable residency status, and mothers who were unemployed or dependent on social welfare or invalidity benefits.

## Discussion

In summary, regardless of region of birth, the vast majority of women in this study had adequate care in terms of timing, frequency of consultations, and self-reported access. Region of birth was an independent risk factor for late initiation of care, but not for infrequent consultations. Women born in Sub-Saharan Africa were more prone to accessing care late but were not more at risk of infrequent consultations. Women born in North Africa, however, had similar adequacy of care compared to the Belgium-born population. The three groups also differed in terms of socioeconomic profiles and socioeconomic predictors of antenatal care use. Housing type, professional activity, and health insurance status were important predictors of both outcomes.

**Table 4. Predictors of late start of antenatal care by region of birth, with Chi² or Fisher exact p-values.**

| Variable | BELGIUM | | | NORTH AFRICA | | | SUB-SAHARAN AFRICA | | |
|---|---|---|---|---|---|---|---|---|---|
| | n | % Late start | p-value | n | % Late start | p-value | n | % Late start | p-value |
| **Age (years)** | | | 0.24** | | | **<0.05*** | | | 0.2* |
| <25 | 53 | 5.7 | | 17 | 0 | | 15 | 6.7 | |
| 25–35 | 278 | 3.6 | | 147 | 2.7 | | 137 | 13.9 | |
| 36+ | 91 | 1.1 | | 99 | 8.1 | | 42 | 19.1 | |
| **Parity** | | | 0.99** | | | 0.50** | | | 0.7* |
| 1 | 186 | 3.2 | | 74 | 2.7 | | 59 | 13.6 | |
| 2–3 | 198 | 3.5 | | 148 | 4.7 | | 94 | 13.8 | |
| 4+ | 38 | 2.6 | | 41 | 7.3 | | 41 | 17.1 | |
| **Education** | | | **<0.001*** | | | 0.87** | | | **<0.01*** |
| Lower secondary or less | 66 | 12.2 | | 99 | 5.1 | | 73 | 23.3 | |
| Upper secondary | 149 | 2.7 | | 104 | 3.9 | | 64 | 12.5 | |
| Higher education | 207 | 1 | | 59 | 5.1 | | 56 | 5.4 | |
| **Single motherhood** | | | **<0.001**** | | | 0.99** | | | **<0.05** |
| No | 376 | 1.9 | | 254 | 4.7 | | 135 | 11.1 | |
| Yes | 46 | 15.2 | | 8 | 0 | | 58 | 22.4 | |
| **Housing type** | | | **<0.001**** | | | 0.59** | | | **<0.001*** |
| Private housing | 375 | 1.6 | | 215 | 4.2 | | 142 | 9.9 | |
| Social housing | 28 | 7.1 | | 37 | 8.1 | | 30 | 13.3 | |
| Not own home | 19 | 31.6 | | 8 | 0 | | 21 | 47.6 | |
| **Professional activity** | | | **<0.01**** | | | **<0.05**** | | | **<0.05*** |
| Employed/student | 305 | 1.6 | | 77 | 0 | | 100 | 9 | |
| Unemployed/social welfare/invalidity | 68 | 8.8 | | 29 | 6.9 | | 40 | 17.5 | |
| None/housewife | 49 | 6.1 | | 156 | 6.4 | | 53 | 22.6 | |
| **Equivalised income** | | | 0.13** | | | 0.39** | | | 0.07 |
| Q1-Q2 (lower) | 149 | 4.7 | | 168 | 3.0 | | 136 | 18.4 | |
| Q3-Q4 (higher) | 265 | 1.9 | | 65 | 4.6 | | 48 | 7 | |
| **Health insurance** | | | 0.07** | | | **<0.05**** | | | **0.01**** |
| Yes | 420 | 3.1 | | 248 | 3.6 | | 161 | 11.2 | |
| No | 2 | 50 | | 14 | 21.4 | | 32 | 31.3 | |
| **Duration of residence** | | | N/A | | | 0.44** | | | 0.11** |
| <5 years | 1 | 0 | | 94 | 3.2 | | 74 | 18.9 | |
| 5–15 years | 3 | 0 | | 125 | 6.4 | | 90 | 8.9 | |
| 16+ years | 418 | 3.4 | | 44 | 2.3 | | 30 | 20 | |
| **Legal status** | | | N/A | | | 0.14** | | | 0.17 |
| Stable | 422 | | | 248 | 4.0 | | 151 | 12.6 | |
| Unstable | 0 | | | 15 | 13.3 | | 43 | 20.9 | |
| **Language** | | | N/A | | | 0.25** | | | 0.47** |
| Fluent | 418 | 3.4 | | 103 | 2.9 | | 128 | 14.1 | |
| Good | 4 | 0 | | 71 | 2.8 | | 49 | 12.2 | |
| With difficulty/not at all | 0 | 0 | | 89 | 7.9 | | 17 | 23.5 | |

* Test for trend.

** Fisher's exact test.

In line with population-based studies carried out in Belgium [8, 27], and with the international literature [10], our data shows that immigrant populations accumulate various dimensions of socioeconomic vulnerability. Interestingly, our results also highlight characteristic

**Table 5. Prevalence of less than recommended ANC visits, with crude and adjusted odds ratios and Wald $X^2$ p-values.**

| | n | % <recomm visits | ORc (95% CI) | p-value | ORa (95% CI) n = 868 | p-value |
|---|---|---|---|---|---|---|
| **Region of Birth** | | | | 0.27 | | 0.06 |
| Belgium | 422 | 19.9 | 1 | | 1 | |
| North Africa | 263 | 16.0 | 0.77 (0.51–1.15) | | 0.75 (0.49–1.13) | |
| Sub-Saharan Africa | 194 | 15.5 | 0.74 (0.47–1.16) | | **0.56 (0.34–0.92)** | |
| **Age (years)** | | | | **<0.001***| | **<0.001*** |
| <25 | 85 | 36.5 | **2.65 (1.62–4.33)** | | **2.78 (1.67–4.63)** | |
| 25–35 | 562 | 17.8 | 1 | | 1 | |
| 36+ | 232 | 10.8 | **0.56 (0.35–0.89)** | | **0.54 (0.33–0.86)** | |
| **Parity** | | | | **<0.001*** | | **<0.001*** |
| 1 | 319 | 37.9 | 1 | | 1 | |
| 2–3 | 440 | 6.4 | **0.11 (0.07–0.17)** | | **0.09 (0.06–0.15)** | |
| 4+ | 120 | 5.8 | **0.10 (0.05–0.22)** | | **0.08 (0.03–0.18)** | |
| **Education** | | | | 0.14 | | 0.08 |
| Lower secondary or less | 238 | 17.7 | 0.82 (0.53–1.3) | | 0.68 (0.43–1.07) | |
| Upper secondary | 317 | 14.8 | 0.66 (0.44–1.00) | | **0.65 (0.43–0.98)** | |
| Higher education | 322 | 20.8 | 1 | | 1 | |
| **Single motherhood** | | | | **<0.001** | | **<0.001** |
| No | 765 | 15.4 | 1 | | 1 | |
| Yes | 112 | 33.9 | **2.82 (1.82–4.4)** | | **2.31 (1.46–3.68)** | |
| **Housing type** | | | | **<0.001** | | **<0.001** |
| Private housing | 732 | 16 | 1 | | 1 | |
| Social housing | 95 | 12.6 | 0.76 (0.40–1.44) | | 0.67 (0.34–1.31) | |
| Not own home | 48 | 54.2 | **6.21 (3.41–11.33)** | | **4.78 (2.51–9.11)** | |
| **Professional activity** | | | | **<0.001** | | **<0.001** |
| Employed/student | 482 | 17.8 | 1 | | 1 | |
| Unemployed/social welfare/invalidity | 137 | 27.7 | **1.78 (1.14–2.75)** | | **1.65 (1.04–2.60)** | |
| None | 258 | 12.4 | **0.65 (0.42–1.01)** | | **0.58 (0.37–0.92)** | |
| **Equivalised household income** | | | | **<0.05** | | 0.07 |
| Q1 (lowest) | 245 | 24.1 | 1.64 (0.97–2.75) | | 1.43 (0.84–2.43) | |
| Q2 | 208 | 15.9 | 0.97 (0.55–1.72) | | 0.86 (0.48–1.53) | |
| Q3 | 219 | 13.7 | 0.82 (0.46–1.46) | | 0.79 (0.44–1.41) | |
| Q4 (highest) | 154 | 16.2 | 1 | | 1 | |
| **Health insurance** | | | | **<0.01** | | **<0.01** |
| Yes | 829 | 16.8 | 1 | | 1 | |
| No | 48 | 35.4 | **2.72 (1.47–5.06)** | | **2.05 (1.06–3.98)** | |
| **Duration of residence** | | | | **<0.05** | | **<0.05** |
| <5 years | 169 | 22.5 | 1.21 (0.79–1.85) | | 1.09 (0.70–1.70) | |
| 5–15 years | 218 | 10.6 | **0.49 (0.30–0.80)** | | **0.46 (0.28–0.75)** | |
| 16+ years | 492 | 19.3 | 1 | | 1 | |
| **Legal status** | | | | **<0.05** | | 0.08 |
| Stable | 821 | 16.8 | 1 | | 1 | |
| Unstable | 58 | 31.0 | **2.23 (1.24–4.00)** | | 1.75 (0.94–3.26) | |
| **Language** | | | | 0.24 | | 0.22 |
| Fluent | 649 | 19.0 | 1 | | 1 | |
| Good | 124 | 12.9 | 0.63 (0.36–1.11) | | 0.64 (0.36–1.13) | |
| With difficulty/not at all | 106 | 16.0 | 0.82 (0.47–1.42) | | 0.75 (0.43–1.33) | |
| **Pregnancy complications** | | | | 0.12 | | 0.07 |

*(Continued)*

**Table 5.** (Continued)

| | n | % <recomm visits | ORc (95% CI) | p-value | ORa (95% CI) n = 868 | p-value |
|---|---|---|---|---|---|---|
| No | 385 | 20.0 | 1 | | 1 | |
| Yes | 494 | 16.0 | 0.76 (0.54–1.1) | | 0.71 (0.50–1.02) | |
| **Gestation at delivery** | | | | 0.32 | | |
| <37 weeks | 51 | 25.5 | 1.66 (0.86–3.20) | | | |
| 37–40 weeks | 707 | 17.1 | 1 | | | |
| 41+ weeks | 110 | 17.3 | 1.01 (0.59–1.72) | | | |
| **Late start of ANC** | | | | **<0.001** | | |
| No | 825 | 16.1 | 1 | | | |
| Yes | 54 | 42.6 | **3.86 (2.18–6.83)** | | | |

Less than recommended number of consultations: <10 consultations for primipara; <7 consultations for multipara.

* Test for trend.

**ORa**: adjusted for late start of ANC (first trimester vs. later) and gestation at delivery (<37 weeks; 37–40 weeks; 41–42 weeks).

differences between the immigrant groups, such as different patterns of professional activity, housing type, lone parenthood, and immigration status, which reflect their specific migration histories, culture, and socioeconomic integration in Belgium.

Region of birth was found to be an independent risk factor for late initiation of care, but not for infrequent consultations. With some exceptions [14, 15, 28], most European studies analysed immigrant women as a single group [13], which has the limitation of masking key differences within. Our results revealed important disparities in terms of follow-up between the two groups. Indeed, women born in North Africa were not particularly at risk of late start nor of insufficient consultations, whereas women born in Sub-Saharan Africa were more likely to initiate care late, compared to native-born women. Once the first contact with healthcare professionals was established, most women from Sub-Saharan Africa appeared to receive adequate follow-up nonetheless, which indicates a responsive antenatal care system, and compliance on the patient's part. It is possible that the high frequency of consultations despite a delayed start of care are the consequence of pregnancy complications which arose due to the lack of antenatal care in the first trimester, although additional analyses of our data offer no support for this claim. In fact, we found that the prevalence of at least one complication was 56% among women who first consulted in the first trimester, and 54% among women who consulted later.

In this respect, part of the excess adverse perinatal health outcomes described in the literature [6] for women born in Sub-Saharan Africa could potentially be due to their higher rates of delayed start of care; this is a hypothesis which should be corroborated by future studies.

Systematic reviews carried out in 2011 and 2013, found that in most included studies, ethnic minority groups or foreign-born women were more likely to initiate care late, to have fewer visits, or to have inadequate care [13, 17]. The particular vulnerability of women from Sub-Saharan African countries in terms of late initiation has been highlighted before [14, 15, 28]. When analysing potential explanatory factors, a French study concluded that women's legal status was a greater determinant than geographic origin, with the higher proportion of inadequate care among migrants from SSA probably reflecting the higher proportion of undocumented women among them [14]. One study carried out in the Netherlands fifteen years ago found that women born in Morocco had three times the prevalence of late initiation of care compared to Dutch women [29]. Adjusting for socioeconomic and health-related factors significantly reduced, but did not eliminate the increased odds. The difference in context,

timing, healthcare system, and possibly migration type are potential explanations for the discrepancy with our results.

We found a series of sociodemographic and socioeconomic factors to be associated with antenatal care, which reflects the strong impact of living conditions, poverty and health literacy on healthcare use, and possibly downplays some of the widely reported effects of ethnicity or country of origin on their own. It is crucial to remember that socioeconomic deprivation can itself be the consequence of immigration, due to the process of social exclusion [10]. We found that insufficient income, education, poor living conditions and more precarious working status were important barriers to adequate use of ANC for women born in Belgium as well as for those born in Sub-Saharan African countries. Similarly, population-based studies in Belgium had found a strong association between socioeconomic factors and perinatal health outcomes such as low birthweight and perinatal mortality in women from these two nationality groups, but not in women from North Africa [6, 27]. The impact of shorter education on healthcare use might be mediated by lower health system literacy, affecting women's competences to access and navigate the healthcare system [30]. For instance, lack of knowledge about the healthcare structure, about eligibility, and existing services were found to be the most prevalent barrier to receiving optimal antenatal care among recent immigrants in Norway [31].

Compared to earlier studies, this is one of the few [14, 15] to include an indicator of housing type, which interestingly, appears to have a particularly strong association with both late initiation of care and infrequent visits. Our results show that the subgroup of women who lack their own accommodation are extremely at risk of having inadequate antenatal care. Secondary analyses (S2 Table) revealed that this group accumulated various socioeconomic risk factors.

The lack of healthcare coverage was a predictor for both outcomes, more strongly for late initiation of care. This indicates that the lack of health insurance is a greater barrier to initiating care than to continuing care. In fact, in Brussels, most women without health insurance at the beginning of their pregnancy are helped by healthcare professionals and collaborating social services once they have had a first contact, to gain access to health insurance, free care, or financial reimbursement, depending on their situation. Sometimes, it is more the lack of information concerning eligibility to care which hampers access, rather than the financial barrier itself. This is consistent with the experiences of some of the undocumented women in our sample who reported not initiating care because they were unaware of having cost-free and confidential access. Without any health insurance the financial costs of care could be prohibitive and the administrative procedure to obtain "Urgent Medical Care" appears unsurmountable for some [32]. Our findings highlight the necessity of ensuring that all women have the rights, the information, and logistic and financial access to a first contact with perinatal healthcare professionals, which would be more straightforward if health insurance were universal.

Regardless of region of birth, the vast majority of women in this sample received adequate antenatal care in terms of timing, frequency of consultations, and self-reported access. The prevalence of late initiation was low (6.1%), and most late starters consulted no later than the fourth pregnancy month. A study carried out in another large Belgian city in 2015 also found that 6% of women initiated care late [33], with foreign-born women being twice as exposed, compared to those born in Belgium. However, two studies carried out in Brussels in 2008 and 2016 found higher estimates. The first study was a prospective observational study of 333 women recruited in clinical centres, and found that 10.8% initiated care after 12 weeks of gestation [34]. The second was a population-based study including 11 380 low-risk pregnancies. This study found that 6.7% of women hadn't consulted by 20 weeks of gestation [23]. Both studies excluded women with comorbidities or high-risk pregnancies, a subgroup which we can expect to be more prone to consulting early [16]. What is more, the second study also noted that the rates of late start were on a downward trend over their six-year study period

(2010–2016), which suggests that the prevalence of late start might have continued to decrease thereafter. Furthermore, it should be remembered that our sample focused on three specific birth regions, which clearly represent a subset of the population and somewhat limits comparability with other studies. The rates of late initiation of ANC appear to be significantly higher in some of our neighbouring countries. Research from the UK, for instance, has estimated this prevalence at more than 30% [15, 35]. In the Netherlands it has been found to be around 10% for Dutch women, and higher for ethnic minorities such as Moroccan (30%) [28, 29]. A study carried out in Norway this year found that 16% of recent immigrants consulted after the first trimester, although only 2.5% initiated care after week 21 [31]. A recent study carried out in France, also found higher estimates of women consulting past the first trimester: 14% for non-migrants, 19% for legal migrants, 30% for undocumented migrants [14]. Thus, in comparison to these observations, the initiation of antenatal care in Brussels appears to be exceptionally good. Increasingly, non-profit organisations in Brussels help pregnant women access healthcare and offer psychosocial support [36–38], which could partly explain the improving and overall adequate antenatal care use. The estimate of late initiation of ANC used in our analyses was based on self-reported timing. If we used the timing extracted from clinical records instead, the prevalence increased to 10%. We can be quite certain that this latter figure is an overestimation given that 29% of mothers had at least part of their follow-up outside of the hospital where the data was extracted, meaning that the first consultation appearing in the hospital software (which we extracted) was not their first one. When we calculated late initiation using clinical data except when women were partly followed-up elsewhere, we found an estimate of 6.7%, which only slightly exceeds the self-reported one. We can thus be rather confident that the true measure lies around 6–7%, and certainly no more than 10%.

Most late starters consulted just a few weeks into their second trimester, and only 15% of them reported having had difficulties getting an appointment. What is more, the main reasons that women gave for initiating care late were the delayed recognition of pregnancy and the lack of perceived need to consult in the first trimester. These findings indicate that the late initiation of care was not predominantly due to access barriers such as lack of legal access, difficulty navigating the system or getting an appointment, but often due to a lack of perceived need. These are certainly not the sole mechanisms, but they do suggest that ensuing courses of action aimed at improving adequacy of ANC and reducing inequalities, should include the following: facilitate women's recognition of early pregnancy; ensure adequate access to pregnancy tests or diagnosis by a clinician; and improve women's (and their partners') awareness of the importance of early initiation of care.

It is conceivable that public health interventions increasing awareness about the importance of starting ANC in the first trimester could shift the initiation of care forward by a few weeks. This would be all the more relevant given that the women most prone to initiating care late appear to be those who might benefit most from a timely start. In fact, antenatal care not only serves to enhance physical health, but is also an entry door into social and psychological support, which can be particularly valuable to women living in socioeconomic deprivation. The fact that delayed initiation of care was the strongest predictor of insufficient consultations is certainly not surprising, but it underlines, yet again, the importance of a timely start.

Overall, very few women revealed having had difficulties getting an appointment (4%) or wishing to have had more visits (7%). However, a high proportion (17.8%) had received less than the 10 visits recommended for primipara and the 7 for multipara. This high prevalence was mainly driven by primipara having attended between 7 and 9 antenatal visits. Whether having one or a few consultations less than what is recommended should be considered "inadequate" is subject to debate; the WHO, for instance, considers 8 antenatal visits to be adequate [39]. Indeed, other studies have used lower thresholds of inadequate care, e.g. less than 50% of

recommended visits [14], no care, or less than 3, 4, or 7 consultations [13]. Furthermore, when we measured the proportion of primipara having attended less than 10 consultations in clinical records, we found it to be slightly lower than that reported by women (10% vs. 14%), thus it is possible that we have slightly overestimated the prevalence of women having had less than the recommended number of visits.

One of the strengths of this study concerns the inclusion of a series of individual-level indicators which cannot be obtained through routine data (e.g., legal status, income, housing type, difficulty accessing ANC, reasons for consulting late). The inclusion of three particular nationality groups was deliberate and has allowed us to explore their specificities. Furthermore, the administration of the questionnaire via face-to-face interviews and the use of interviewers speaking the most prevalent languages allowed us to obtain a high participation rate, minimising selection bias and missing data.

One limitation of this study is that there were few participants in the most at-risk categories, such as women without their own accommodation or women without health insurance. This, together with the low prevalence of late initiation of care, limited the statistical power, yielding large confidence intervals and preventing us to carry out multivariable models stratified by country of birth. Future studies purposefully including larger numbers of women with the most precarious profiles could overcome these limitations. Furthermore, despite our efforts to avoid excluding mothers, we were not able to include roughly 1% due to language barriers, and 1.8% who were too unwell to participate (due to illness or exhaustion). However, these low percentages preclude any significant impact on the generalisability of the study. We must also recognise that in the same way that analysing region of birth groups separately enabled us to discern important differences, we could be missing further crucial variations within each of these groups. For instance, Sub-Saharan African countries include a vast spectrum of realities in terms of socioeconomic contexts, culture, and types of migration.

In order to assess the generalisability of this study, we have compared some of the sociodemographic characteristics of the included participants with complete population data of women having given birth in Brussels between 2005–2010 [27]. Bearing in mind that the population data precedes our data collection, and that it is limited to singleton births, we find that the relative differences between the three groups are comparable (e.g. highest proportion of primipara among women born in Belgium, followed by women born in Sub-Saharan Africa; highest percentage of single motherhood among Sub-Saharan African mothers and lowest among North African; proportion of women with university degree highest among women born in Belgium and lowest among those born in North-Africa). However, in our sample we included fewer very young mothers and more highly educated mothers, compared to the overall birth population. Although this tendency is common among research study participants, it somewhat limits the generalisability of the study.

Finally, we would like to underline the existence of numerous models of antenatal care, and the lack of agreement on best practice, including care provider type, number of visits and continuity of care [40, 41]. In this study, we have relied on the national Belgian guidelines to assess the adequacy of initiation and frequency of care. Furter research to identify the most adequate antenatal care pathways would be valuable, not only to improve maternal and neonatal outcomes, but also to enable meaningful international comparisons on healthcare use.

## Conclusions

The large majority of women in this study had adequate care in terms of timing, frequency of consultations, and self-reported access, regardless of their country of origin. In most cases,

delayed start of care was due to a lack of perceived need, rather than to administrative, financial, or logistic access barriers.

The three region of birth groups differed in terms of socioeconomic profiles, use of antenatal care, and in terms of socioeconomic predictors of ANC use. Women born in Sub-Saharan Africa were more prone to accessing care late but were not more at risk of having infrequent consultations. Women born in North Africa, on the other hand, had similar adequacy of care compared to the Belgium-born population.

The inequalities in ANC use were mainly related to few but highly discriminating socioeconomic determinants, such as housing type, professional activity, and health insurance status.

Future research into migrants' use of ANC should therefore continue to take into account their place of birth and socioeconomic characteristics. Furthermore, additional dimensions of adequacy of care, such as quality and content of care, could provide useful additional insights.

Based on our findings, we recommend three types of policies to improve equitable access and utilisation of healthcare: 1) increasing early uptake of ANC by improving timely recognition and testing of pregnancy, and by raising awareness on the importance of care in the first trimester, 2) improving immigrant women's socioeconomic integration, and 3) universalising the access to public health insurance, which in Belgium would mean removing the logistic barrier of applying for "Urgent Medical Care".

## Supporting information

**S1 Table. Region of birth classification.**
(XLSX)

**S2 Table. Participant characteristics by housing type.**
(XLSX)

**S1 File. Adapted migrant friendly maternity care questionnaire.** French version.
(PDF)

**S2 File. Adapted migrant friendly maternity care questionnaire.** English version.
(PDF)

## Acknowledgments

We express our gratitude to all study participants, and to the heads of obstetrics and neonatal services of CHU Brugmann, CHU St Pierre, Erasme and Iris-Sud Ixelles hospitals for giving us permission to carry out the study. We also thank hospital staff (midwives and gynaecologists) for the logistic support in recruiting mothers and accessing clinical data. We are also grateful to the members of the steering committee, including Dr Barlow and Professor Godin, for guiding the design of this research. We would also like to thank the interviewers.

## Author Contributions

**Conceptualization:** Claudia Schönborn, Katia Castetbon, Myriam De Spiegelaere.

**Data curation:** Claudia Schönborn.

**Formal analysis:** Claudia Schönborn.

**Funding acquisition:** Claudia Schönborn, Myriam De Spiegelaere.

**Investigation:** Claudia Schönborn.

**Methodology:** Claudia Schönborn, Katia Castetbon, Myriam De Spiegelaere.

**Project administration:** Claudia Schönborn.

**Software:** Claudia Schönborn.

**Supervision:** Katia Castetbon, Myriam De Spiegelaere.

**Visualization:** Claudia Schönborn.

**Writing – original draft:** Claudia Schönborn.

**Writing – review & editing:** Katia Castetbon, Myriam De Spiegelaere.

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
