## [Decision Letter · Decision Letter 0]

18 Nov 2021

PONE-D-21-15143Country of birth as a potential determinant of inadequate antenatal care use among women giving birth in Brussels. A cross-sectional study.PLOS ONE

Dear Dr. Schönborn,

Thank you for submitting your manuscript to PLOS ONE. After careful consideration, we feel that it has merit but does not fully meet PLOS ONE’s publication criteria as it currently stands. Therefore, we invite you to submit a revised version of the manuscript that addresses the points raised during the review process.

We look forward to receiving your revised manuscript.

Kind regards,

Angela Lupattelli, PhD

Academic Editor

PLOS ONE

Journal Requirements:

2. Please include additional information regarding the survey or questionnaire used in the study and ensure that you have provided sufficient details that others could replicate the analyses. For instance, if you developed a questionnaire as part of this study and it is not under a copyright more restrictive than CC-BY, please include a copy, in both the original language and English, as Supporting Information. Moreover, please include more details on how the questionnaire was pre-tested.

3. Please state in your methods section whether you obtained consent from parents or guardians of the minors included in the study or whether the research ethics committee or IRB approved the lack of parent or guardian consent.

"This work was supported by an FNRS (Fund for Scientific Research https://www. frsfnrs. be/ en/) FRESH doctoral grant for the first author (CS), and a FER (Fonds d’Encouragement à la Recherche) grant from the Université libre de Bruxelles (https://www.ulb.be/en/research) to carry out the data collection (MDS). 

We note that one or more of the authors is affiliated with the funding organization, indicating the funder may have had some role in the design, data collection, analysis or preparation of your manuscript for publication; in other words, the funder played an indirect role through the participation of the co-authors. If the funding organization did not play a role in the study design, data collection and analysis, decision to publish, or preparation of the manuscript and only provided financial support in the form of authors' salaries and/or research materials, please do the following:

a. Review your statements relating to the author contributions, and ensure you have specifically and accurately indicated the role(s) that these authors had in your study. These amendments should be made in the online form.

b. Confirm in your cover letter that you agree with the following statement, and we will change the online submission form on your behalf: 

“The funder provided support in the form of salaries for authors [insert relevant initials], but did not have any additional role in the study design, data collection and analysis, decision to publish, or preparation of the manuscript. The specific roles of these authors are articulated in the ‘author contributions’ section.

Reviewers' comments:

Reviewer's Responses to Questions

**Comments to the Author**

1. Is the manuscript technically sound, and do the data support the conclusions?

Reviewer #1: Yes

Reviewer #2: Partly

2. Has the statistical analysis been performed appropriately and rigorously? 

Reviewer #1: Yes

Reviewer #2: No

3. Have the authors made all data underlying the findings in their manuscript fully available?

Reviewer #1: Yes

Reviewer #2: No

4. Is the manuscript presented in an intelligible fashion and written in standard English?

Reviewer #1: Yes

Reviewer #2: No

5. Review Comments to the Author

Reviewer #1: I do not have that many comments. It is a well presented project. I am familiar with the used instrument. My main concern is to make the paper more "international": there are so many models of maternity care and antenatal care, and no real agreement on policy, care provider type, number of visits and continuity of care. The most general ones are the WHO guidelines (8 visits for example). GP, OB/GYN, Midwives, nurses, nurse-MW and auxiliary personnel deliver babies and conduct ANC. I would describe this in general, and also present the special case of system in Brussels. You measure if the mothers comply with the Belgian system, and thus, you measure also health system literacy. You should also say more about financial implications; insurance in general for MCH services. So line 118 etc on page 6 could come in the intro.

Methods: I would have liked to have the variables presented first, before the data sets and data collection, for clarity.

So a good study. Finding comparable to those of Denmark (Villadsen, Mamaact) and Norway (Bains, Mipreg), and probably more.

Reviewer #2: The paper investigates the associations between maternal birth regions and with the use of antenatal care in Brussels, an important issue that needs to be taken into account when developing antenatal care for migrant women. The paper includes important information about delayed start of antenatal care (ANC). However, the outcome of number of visits is more problematic when gestational week is not considered. Visits as recommended would probably indicate some complications for women with a late start in ANC having shorter time period for the visits. The paper might benefit from focusing on late start in ANC only. A suggestion is to just report shortly the number of visits in relation to region of birth in order to problematize the issue in the discussion.

The paper needs a thorough revision to be clear in terms of definitions and concepts. For example, what countries are included in the regions of North Africa and and Sub-Saharan Africa? This is not defined in the method section where you refer to UN definition. In the abstract, North Africa is also described as Maghreb. I get a bit confused when inadequate antenatal care is synonymously with late start and number of visits rather than content of care.

These are among the things that need to be clarified:

Abstract

The abstract needs to be revised so that study details are clear for example:

Aim: the use of antenatal care needs to be defined

Clinical data was also collected. From patient records?

Outcomes should be defined.

Results should be revised and results in terms of numbers, %, OR and CI 95% should be reported for the two outcomes.

Conclusion could be improved according to the study objective.

Background

Why timely start in ANC is important could be further described. The language could be improved and concepts clarified and defined. For instance the effects of migration on perinatal health could be changed to the impact of, perinatal mortality would be in infants to North and Sob-Saharan African mothers and women from Maghreb were protected for this outcome. What is meant by protected?

Please also use one term for North Africa / Maghreb throughout the manuscript.

R 56. prevent diseases – prevent pregnancy complications would be more appropriate?

Methods

R 89. To what population could you generalise the results to? A selection bias has probably been introduced by a selection of hospitals with the highest proportions of women with an African nationality, two with more socially disadvantaged patients and two with more affluent ones.

The rational to create a full model with a range of socioeconomic factors could be developed and discussed. What remains in the exposure when these variables are adjusted for would be interesting to get the authors’ perspective on?

R92. Please include what countries of birth that is included in the respective region.

R96. Too unwell? For what? Please specify and discuss in relation to generability.

R98. How the questionnaires were developed and translated is somewhat unclear. The final version (?) was translated into English, and orally into Moroccan dialect. Translation into other languages was done ad hoc. Please develop so that the reader may understand the procedure, the questionnaires, the translations and in what languages the questions were translated ad hoc. How did this affect the results?

R101: Information collected from hospital records: How was this information transferred from the ANC? Are the measures valid?

R106. Consent was sought, but also received?

R114 Validity of self-reported timing of initiation of care. When was it measured?

Do women rembember how many visits they make? Recall bias here?

R121 When women commence in ANC late, for instance in gwk 25 or 28, how many visits are then recommended in futher ANC? What is adequate care then?

Measurements are decribed in a relatively large section and could probably be condensed.

Income: there seem to be a flaw in the income categories: for instance 1000-1500 and 1500-2000,

Statistics

Step-by-step backward elimination of indicators may be questioned. Suggest consult statistician here.

Results

A large majority (34%) among the Sub-Saharan are not described in relation to country of birth. If they are from Somalia this will mean that they are at a much higher risk for adverse outcomes. Please add information.

Discussion

A short summery of results would make the discussion more easy to read.

R355 Do you have information on all undocumented migrants or is there a selection bias here?

Very few references in the discussion

I get a bit confused over some concepts when reading the discussion for example adequate care which would be much wider than the definition in this paper. For example:

R381. The fact that delayed initiation of care was the strongest predictor of insufficient consultations is certainly not surprising, but it underlines, yet again, the importance of a timely start.

6. PLOS authors have the option to publish the peer review history of their article (what does this mean?). If published, this will include your full peer review and any attached files.

Reviewer #1: **Yes: **Johanne Sommerschild Sundby

Reviewer #2: No

---

## [Author Response · Author response to Decision Letter 0]

18 Mar 2022

On behalf of all co-authours, we would like to thank the reviewers and editors for their interest in this manuscript and for all suggestions which allowed us to significantly improve it. We have replied to all questions and comments point by point in the word document entitled "Response to reviewers". 

We include a copy of all responses below, although we would strongly advise reading the word document instead for better legibility and visualisation of tables.

With Kind Regards,

Claudia Schönborn

Journal Requirements:

Thank you, we have made the relevant amendments (table format, supporting Information, contributions statement).

Please note that the Line numbers refer to the manuscript with track changes.

2. Please include additional information regarding the survey or questionnaire used in the study and ensure that you have provided sufficient details that others could replicate the analyses. For instance, if you developed a questionnaire as part of this study and it is not under a copyright more restrictive than CC-BY, please include a copy, in both the original language and English, as Supporting Information. Moreover, please include more details on how the questionnaire was pre-tested.

We have now included a copy of the questionnaire in French and in English as Supporting Information (referenced in Lines 104-106 of the manuscript with track changes). 

We have also added more information concerning the survey. Further details, including questionnaire development and pilot, have been described in depth and published in the referenced protocol: 

Schönborn C, Castetbon K, Sow M, Racape J, De Spiegelaere M. Mothers’ experiences of perinatal care in Belgian public hospitals: exploring the social inequalities. Protocol for a cross-sectional survey. BMJ Open. 2020;10(11):e038400. doi: 10.1136/bmjopen-2020-038400.

We hope that this meets your requirements for study replicability.

3. Please state in your methods section whether you obtained consent from parents or guardians of the minors included in the study or whether the research ethics committee or IRB approved the lack of parent or guardian consent.

Thank you for this suggestion. We have added this information to the methods section.

"This work was supported by an FNRS (Fund for Scientific Research https://www. frsfnrs. be/ en/) FRESH doctoral grant for the first author (CS), and a FER (Fonds d’Encouragement à la Recherche) grant from the Université libre de Bruxelles (https://www.ulb.be/en/research) to carry out the data collection (MDS). 

We note that one or more of the authors is affiliated with the funding organization, indicating the funder may have had some role in the design, data collection, analysis or preparation of your manuscript for publication; in other words, the funder played an indirect role through the participation of the co-authors. If the funding organization did not play a role in the study design, data collection and analysis, decision to publish, or preparation of the manuscript and only provided financial support in the form of authors' salaries and/or research materials, please do the following:

a. Review your statements relating to the author contributions, and ensure you have specifically and accurately indicated the role(s) that these authors had in your study. These amendments should be made in the online form.

We have reviewed the contributions statement and confirm that it is correct.

b. Confirm in your cover letter that you agree with the following statement, and we will change the online submission form on your behalf: 

“The funder provided support in the form of salaries for authors [insert relevant initials], but did not have any additional role in the study design, data collection and analysis, decision to publish, or preparation of the manuscript. The specific roles of these authors are articulated in the ‘author contributions’ section.

Thank you. We have specified that we agree with this statement in the cover letter. 

Thank you for pointing this out. 

The sections you refer to includes very few, yet informative, findings of our survey. 

Lines 255-260: We thought that an additional table would not be appropriate for presenting these three percentages and we have therefore now stated them in the text. Please let us know, should you consider a separate table to be more appropriate.

Lines 265-267: we have included the relevant data in the text.

Line 399: we have added a Supporting Information Table.

6. We note that you have indicated that data from this study are restricted due to ethical concerns. If data are ethically restricted, we ask the author make the data available upon request to interested and qualified researchers. For more information on unacceptable data access restrictions, please see https://eur01.safelinks.protection.outlook.com/?url=http%3A%2F%2Fjournals.plos.org%2Fplosone%2Fs%2Fdata-availability%23loc-unacceptable-data-access-restrictions&data=04%7C01%7Cclaudia.schoenborn%40ulb.be%7C4d560cd0047e4f69f41608d9df54197f%7C30a5145e75bd4212bb028ff9c0ea4ae9%7C0%7C0%7C637786375308141500%7CUnknown%7CTWFpbGZsb3d8eyJWIjoiMC4wLjAwMDAiLCJQIjoiV2luMzIiLCJBTiI6Ik1haWwiLCJXVCI6Mn0%3D%7C3000&sdata=UXAczriyEC1kkC4mggDi1b73ewo2%2FUVXVyQLyb1NX2U%3D&reserved=0.

a) If there are ethical or legal restrictions on sharing a de-identified data set, please explain them in detail (e.g., data contain potentially identifying or sensitive patient information, data are owned by a third-party organization, etc.) and who has imposed them (e.g., a Research Ethics Committee or Institutional Review Board, etc.). Please also provide non-author contact information for a data access committee, ethics committee, or other institutional body to which data requests may be sent.

We unfortunately cannot publicly share our dataset due to ethical restrictions. In fact, our dataset contains sensitive data, such as sex (the study is restricted to women), age, country of birth, nationality, legal status, delivery hospital and delivery year, detailed clinical data concerning the mother and their child, detailed socioeconomic data such as health insurance status, social welfare status, professional activity etc. Although not directly identifying, in combination these data are likely to become identifying. The Research Ethics Committee has imposed this restriction.

However, we are happy to make a portion of the database available upon reasonable request to interested and qualified researchers.

Data requests may be sent to the principal investigator (CS) or to the Data Protection Unit: 

Data Protection Unit, 

Université libre de Bruxelles, 

Campus du Solbosch - CP 130

Avenue F.D. Roosevelt, 50, 

1050 Bruxelles, 

e-mail : rgpd@ulb.be

 

Reviewer #1: 

I do not have that many comments. It is a well presented project. I am familiar with the used instrument. 

Dear Dr Sommerschild Sundby, we would like to thank you for your interest, for your time in reviewing this article, and for your positive evaluation.

Please note that all line numbers we have referenced concern the manuscript with track changes.

1. My main concern is to make the paper more "international": there are so many models of maternity care and antenatal care, and no real agreement on policy, care provider type, number of visits and continuity of care. The most general ones are the WHO guidelines (8 visits for example). GP, OB/GYN, Midwives, nurses, nurse-MW and auxiliary personnel deliver babies and conduct ANC. I would describe this in general, and also present the special case of system in Brussels. 

Thank you for this comment. We acknowledge the lack of consensus concerning an internationally recognised adequate model of ANC, and have now added this aspect to our manuscript (Lines 508-513). 

Our background and discussion sections provided comparisons and discussions of the international literature, including studies from the UK, Netherlands, and France, and various systematic reviews including studies from a range of industrialised countries (North America & Europe). In Lines 468-472, we examined our results in light of other thresholds of adequacy of care (including the WHO guidelines). In this reviewed version of the article we have develop the discussion and comparisons further. Furthermore, in the methods section, we have now added a paragraph on the specific Belgian “setting”, in which we describe who provides ANC in Belgium, in what setting, and how care can be financially accessed. We hope that you consider this to be adequate.

2. You measure if the mothers comply with the Belgian system, and thus, you measure also health system literacy. 

Thank you for this insight. We recognise that health system literacy might be a contributing factor causing women to access timely ANC and to attend the offered appointments, and we have now considered this in Lines 391-395. 

3. You should also say more about financial implications; insurance in general for MCH services. So line 118 etc on page 6 could come in the intro.

Thank you for this suggestion. Our “setting” section now includes a description of the financial access to ANC in Belgium (Lines 120-133). We agree that financial implications can constitute important barriers to care, so it comes as no surprise in our study that health insurance status was significantly associated with both outcomes. Sometimes, it is more the lack of information concerning eligibility to care which hampers access, rather than the financial barrier itself. We have discussed and further developed these aspects in Lines 401-413. Furthermore, we have considered the importance of financial access to care in our recommendations (Lines 533-534). 

4. Methods: I would have liked to have the variables presented first, before the data sets and data collection, for clarity.

As recommended by PLOS ONE, we have followed the STROBE guidelines for the reporting of cross-sectional studies (https://www.equator-network.org/wp-content/uploads/2015/10/STROBE_checklist_v4_cross-sectional.pdf ), in which study design, setting, and participants come before describing the variables. We would therefore prefer to continue using this structure.

So a good study. Finding comparable to those of Denmark (Villadsen, Mamaact) and Norway (Bains, Mipreg), and probably more.

Thank you, we have updated the paper by including references to articles published since our first submission (Lines 393-395; 432-433).

  

Reviewer #2: 

The paper investigates the associations between maternal birth regions and with the use of antenatal care in Brussels, an important issue that needs to be taken into account when developing antenatal care for migrant women. The paper includes important information about delayed start of antenatal care (ANC). 

We would like to thank you for your revision and valuable comments. We have addressed your questions and suggestions point by point below.

Please note that all line numbers we have referenced concern the manuscript with track changes.

1. However, the outcome of number of visits is more problematic when gestational week is not considered. Visits as recommended would probably indicate some complications for women with a late start in ANC having shorter time period for the visits. The paper might benefit from focusing on late start in ANC only. A suggestion is to just report shortly the number of visits in relation to region of birth in order to problematize the issue in the discussion.

Thank you for this consideration, which raises 3 issues: (1) number of outcomes; (2) inclusion of gestational weeks; (3) and number of visits for women with a late start.

Concerning the first point, we think that the timing of first consultation and the number of visits are two complementary dimensions which, taken together, allow for a more comprehensive understanding of women’s access and use of antenatal care. In fact, an early start of ANC does not necessarily ensure adequate frequency of consultations and vice versa. Our study has highlighted that the socioeconomic and migration-related risk factors for delayed initiation of care are different to those of infrequent consultations. For instance, region of birth was associated with late initiation of care, but not frequency of consultations. Older women were more likely to start ANC late, whereas younger women were more likely to have less than the recommended number of consultations. Moreover, women with a higher parity were more represented among late starters, whereas primipara were more represented in the group with lower consultations. Furthermore, we believe that it is informative to identify the profiles of women having infrequent consultations despite an adequate start of care, and who would, in that respect, have been considered as having had adequate antenatal care. For all these reasons, we would advocate for keeping both outcome measures.

Secondly, you suggest considering gestation at delivery (end of ANC) when analysing the number of consultations. We thank you and agree with this suggestion. Indeed, one would ideally measure the exact number of ANC consultations and compare the value against a guideline specific to the length of pregnancy (start of ANC, and end of ANC secondary to delivery). However, the official Belgian guidelines do not offer a schedule depending on timing of initiation of care or gestational week at delivery; moreover, we were not always able to have an exact number of consultations and had to work with ranges instead.

In order to appreciate how much these elements are related, in Table 5, we have now provided ORs adjusted for initiation of care (1st trimester vs later) and gestation at delivery (<37; 37-40; 41-42 weeks). These results do not differ substantially from the unadjusted ones.

Thirdly, you suggested that “Visits as recommended would probably indicate some complications for women with a late start in ANC having shorter time period for the visits”. Although our data are not supportive of this hypothesis (see below), we agree with your reasoning and have now discussed this possibility in the discussion (Lines 360-365).

The table below shows that there is no difference in the proportions of pregnancy complications in the adequate initiation group versus the delayed initiation group:

Initiation of care % pregnancy complications Chi² p-value

 First trimester (n=825) 56.36 0.7

 Later (n=54) 53.7 

2. The paper needs a thorough revision to be clear in terms of definitions and concepts. For example, what countries are included in the regions of North Africa and and Sub-Saharan Africa? This is not defined in the method section where you refer to UN definition. 

We used the official UN classification of countries to form our region of birth groups and referenced this in the methods section. As the reviewer is considering it more appropriate, we will additionally provide the list as Supplementary Data.

3. In the abstract, North Africa is also described as Maghreb.

 Thank you for spotting this. In line with your comment further down (Q13), we have now changed the region name from Maghreb to North Africa throughout the manuscript. 

4. I get a bit confused when inadequate antenatal care is synonymously with late start and number of visits rather than content of care.

We agree with you that these concepts are not synonymous. We consider late start of care and inadequate number of visits to be two dimensions of inadequate antenatal care. We have made a few amendments to the text in order to clarify even further that we are measuring ANC use (e.g. conclusions Line 525), and not adequacy of care in all its facets (i.e. quality and content of care). This has now also been acknowledged in the abstract and general conclusions (Lines 37-38; 528-529).

 

5. These are among the things that need to be clarified:

Abstract

The abstract needs to be revised so that study details are clear for example:

Aim: the use of antenatal care needs to be defined

The use of antenatal care has been specified in the methods section of the abstract: “a) late start of antenatal care and b) less than minimum recommended number of consultations”. Because the abstract word limit is set at 300 words, it is not possible to define these variables further in this section. Details of these measurements have been provided in the main methods section.

6. Clinical data was also collected. From patient records?

That is correct. We have now added this information to the abstract.

7. Outcomes should be defined.

The outcomes have been specified in the methods section of the abstract (Lines 26-27): “[…] to estimate the associations of socioeconomic and migration characteristics with two dimensions of adequacy of care: a) late start of antenatal care and b) less than minimum recommended number of consultations.” We unfortunately do not have additional space to define these outcomes further in the abstract. 

8. Results should be revised and results in terms of numbers, %, OR and CI 95% should be reported for the two outcomes.

Thank you for this suggestion. We have added the precise estimates for the most relevant results. 

9. Conclusion could be improved according to the study objective.

Thank you for the suggestion; the conclusions have been adapted (Lines 36-39). 

10. Background

Why timely start in ANC is important could be further described. 

The importance of timely start of antenatal care has been further developed in the background section (Lines 61-64).

11. The language could be improved and concepts clarified and defined. For instance the effects of migration on perinatal health could be changed to the impact of, 

We have changed “effects” to “impact” (Line 47).

12. perinatal mortality would be in infants to North and Sob-Saharan African mothers and women from Maghreb were protected for this outcome. What is meant by protected?

By protected, we meant that women with a North African nationality were less at risk of having a premature delivery, compared to the reference group. We have clarified this in Lines 54-55.

13. Please also use one term for North Africa / Maghreb throughout the manuscript.

Thank you for your suggestion. We have now modified the term to North Africa throughout.

14. R 56. prevent diseases – prevent pregnancy complications would be more appropriate?

This has been modified.

15. Methods

R 89. To what population could you generalise the results to? A selection bias has probably been introduced by a selection of hospitals with the highest proportions of women with an African nationality, two with more socially disadvantaged patients and two with more affluent ones.

We have chosen hospitals with the highest births to North and Sub-Saharan African mothers in order to include sufficient numbers of mothers with immigrant background. Furthermore, in order to include socioeconomic diversity we have chosen hospitals with more and less precarious catchment populations. 

We have now added a reflection on the generalisability of our results in the discussion (Lines 497-507). In fact, “we have compared some of the sociodemographic characteristics of the included participants with complete population data of women having given birth in Brussels between 2005-2010 (Sow et al. 2018). Bearing in mind that the population data precedes our data collection, and that it is limited to singleton births, we find that the relative differences between the three groups are comparable (e.g. highest proportion of primipara among women born in Belgium, followed by women born in Sub-Saharan Africa; highest percentage of single motherhood among Sub-Saharan African mothers and lowest among North African; proportion of women with university degree (or equivalent) highest among women born in Belgium and lowest among those born in North-Africa). However, in our sample we find fewer very young and more highly educated mothers, compared to the overall population of mothers. Although this tendency is common among research study participants, it somewhat limits the generalisability of the study.” 

Reference:

Sow M, Racape J, Schoenborn C, De Spiegelaere M. Is the socioeconomic status of immigrant mothers in Brussels relevant to predict their risk of adverse pregnancy outcomes? BMC Pregnancy and Childbirth. 2018;18(1). doi: 10.1186/s12884-018-2043-3.

16. The rational to create a full model with a range of socioeconomic factors could be developed and discussed. What remains in the exposure when these variables are adjusted for would be interesting to get the authors’ perspective on?

Thank you for this consideration. In the original version of this article we had chosen to create a full model and then, in an exploratory approach, to identify and retain only the most significant predictors. However, we acknowledge the limitations of proceeding via step-by-step removal of predictors based on their p-values (as you also questioned in Q27). This is especially true for studies that, like ours, include rare categories, which limits the possibility of achieving statistical significance at the 0.05 level. In fact, indicators with rare categories might well influence the outcome (or be associated with the outcome) but due to the small sample size may appear as statistically not significant at the 0.05 level. Furthermore, it is informative to measure the effects of each predictor, adjusted for the other socioeconomic characteristics. Although the results do not change significantly, we believe that it is a better approach. We have therefore modified the table to show the full model instead. 

For your information, we include below the comparison of the full and reduced models: see "Response to reviewers word doc".

R92. Please include what countries of birth that is included in the respective region.

This has now been done (see also answer to Q2).

17. R96. Too unwell? For what? Please specify and discuss in relation to generability.

Our interviewers always got the midwives’ permission before approaching the eligible patients to explain the study and invite them to participate. On a few rare occasions, the midwives asked that a patient not be approached on that day (if they were ill, e.g. from pneumonia or infection, exhausted, or psychologically unwell); we have clarified this in the text (Lines 491-493). A few of these women were included in the study a few days later when they were feeling better. Out of the 1079 eligible women, 19 were not included due to being unwell (1.8%).

This is unlikely to have significantly impacted the results given the low occurrence and the fact that the women who were unwell were not differentially distributed in terms of nationality.

18. R98. How the questionnaires were developed and translated is somewhat unclear. The final version (?) was translated into English, and orally into Moroccan dialect. Translation into other languages was done ad hoc. Please develop so that the reader may understand the procedure, the questionnaires, the translations and in what languages the questions were translated ad hoc. How did this affect the results?

The adaptation and translation of the questionnaire have been comprehensively described in the published protocol, which is referenced in this manuscript. The “final version” (Line 107) referred to the final version of the questionnaire after modification of the original MFMCQ. Given that this process has not been detailed in this article we have now adapted the expression.

Following your suggestion, in the methods section (Lines 107-109) we have added information concerning the languages in which the questionnaires were administered: “Nearly all questionnaires were administered in French (86%) and Arabic (13%). Very few questionnaires were administered in English (n=4), or translated ad hoc into Peul (n=2), Riff (n=3), or Dutch (n=2)”.

19. R101: Information collected from hospital records: How was this information transferred from the ANC? Are the measures valid?

Almost all women were at least partly followed up during pregnancy in the hospital where they delivered, so we collected the clinical information directly in the hospital/ANC computer program. Only 5 women had no ANC, so no information needed to be collected in these cases.

For women fully followed-up in the hospital where they delivered, the clinical information was valid (i.e. information directly copied from medical notes).

In the case of women who had initially or partly been followed-up elsewhere (29%), this information is more prone to some imprecision. This is why we collected outcome data both from hospital records and from participant interviews, which allowed us to triangulate the information. These aspects have been discussed in Lines 439-446.

20. R106. Consent was sought, but also received?

That’s correct. We have modified the expression (Line 117).

21. R114 Validity of self-reported timing of initiation of care. When was it measured?

Given that it is a cross-sectional study, all information was collected at the same time (usually a couple of days after birth) - see methods section. Timing of initiation of care was measured through both self-report by participants and extraction from hospital records on the day the interview was carried out. The validity is now discussed in Lines 439-446.

22. Do women rembember how many visits they make? Recall bias here?

The number of consultations was measured both in hospital records and through the questionnaire. These measures were compared to each other and the reliability of such an approach is discussed in the text (Lines 472-475): “when we measured the proportion of primipara having attended less than 10 consultations in clinical records, we found it to be slightly lower than that reported by women (10% vs. 14%), thus it is possible that we have slightly overestimated the prevalence of women having had less than the recommended number of visits”. 

Recall bias would have occurred if women of a certain origin or socioeconomic profile had been more prone to remembering a higher (or lower) frequency of consultations; as far as we know, there are no data to support this assumption.

23. R121 When women commence in ANC late, for instance in gwk 25 or 28, how many visits are then recommended in futher ANC? What is adequate care then?

This is an interesting question. The official Belgian guidelines do not offer this information. The recommendation is of an overall number of visits for uncomplicated pregnancies, differentiating primipara from women having already given birth. Adequate care in the case of delayed initiation will have to be defined by the responsible healthcare professionals in the given context. 

As discussed in the answer to Q1, we have now provided estimates adjusted for delayed initiation of care and gestation at delivery, in order to account for the lower number of visits.

24. Measurements are decribed in a relatively large section and could probably be condensed.

We consider that describing our variables in detail is important for clarity, interpretation, and reproducibility of the study. We would therefore prefer to keep it as it is.

25. Income: there seem to be a flaw in the income categories: for instance 1000-1500 and 1500-2000,

We asked women about their net monthly household income and provided ranges of values both to encourage answering the question and to aid them to provide an estimate, because giving an exact figure is not easy. Furthermore, monthly incomes tend to vary somewhat from month to month and are unlikely to be a perfectly round figure (e.g. 1500.00€). The category 1000-1500 is interpreted as 1000.01€-1500.00, the next 1500.01-2000.00€. The participants were always able to classify their income in one of the provided answer options.

26. Statistics

Step-by-step backward elimination of indicators may be questioned. Suggest consult statistician here.

Thank you for your suggestion. We have updated this, as explained in the answer to Q16.

27. Results

A large majority (34%) among the Sub-Saharan are not described in relation to country of birth. If they are from Somalia this will mean that they are at a much higher risk for adverse outcomes. Please add information.

We have amended the text (Lines 234-236) in order to provide a more detailed description of Sub-Saharan African countries. To avoid a long list of countries with tiny percentages, we chose to name only the countries most represented in this sample: “There were 20 Sub-Saharan African countries of birth, with the most represented being the Democratic Republic of Congo (31.4%), Guinea (19.6%), and Cameroon (16.0%), followed by Rwanda (5.2%), Senegal (4.7%), and Côte d’Ivoire (4.2%).” The remaining 14 countries included less than 3% of mothers in each group.

28. Discussion

A short summary of results would make the discussion more easy to read.

Thank you for this suggestion, a brief summary has now been added (Lines 338-345).

29. R355 Do you have information on all undocumented migrants or is there a selection bias here?

Within our sample, there is no missing information concerning undocumented women (n=25). It should be remembered, however, that our sample only included three birth regions (Belgium, North Africa, Sub-Saharan Africa), so undocumented (or documented for the matter) immigrants from other world regions are not represented in this article. As far as we can tell, there is no selection bias concerning undocumented migrants.

30. Very few references in the discussion

We had initially referenced 20 articles in the discussion, and we have added more as part of the revision process. We have interpreted our findings in the light of the national and international literature, and consider this number of references to be common practice and appropriate for a primary research study.

 

31. I get a bit confused over some concepts when reading the discussion for example adequate care which would be much wider than the definition in this paper. For example:

R381. The fact that delayed initiation of care was the strongest predictor of insufficient consultations is certainly not surprising, but it underlines, yet again, the importance of a timely start.

As also discussed in the answer to Q4, we have now clarified the fact that initiation of care and frequency of consultations are two non-exhaustive aspects of adequacy of care (e.g. Lines 37-38; 528-529). We hope that this avoids any confusion.

---

## [Decision Letter · Decision Letter 1]

4 Apr 2022

Country of birth as a potential determinant of inadequate antenatal care use among women giving birth in Brussels. A cross-sectional study.

PONE-D-21-15143R1

Dear Dr. Schönborn,

We’re pleased to inform you that your manuscript has been judged scientifically suitable for publication and will be formally accepted for publication once it meets all outstanding technical requirements.

Kind regards,

Angela Lupattelli, PhD

Academic Editor

PLOS ONE

Reviewers' comments:

Reviewer's Responses to Questions

**Comments to the Author**

1. If the authors have adequately addressed your comments raised in a previous round of review and you feel that this manuscript is now acceptable for publication, you may indicate that here to bypass the “Comments to the Author” section, enter your conflict of interest statement in the “Confidential to Editor” section, and submit your "Accept" recommendation.

Reviewer #1: All comments have been addressed

2. Is the manuscript technically sound, and do the data support the conclusions?

Reviewer #1: Yes

3. Has the statistical analysis been performed appropriately and rigorously? 

Reviewer #1: Yes

4. Have the authors made all data underlying the findings in their manuscript fully available?

Reviewer #1: Yes

5. Is the manuscript presented in an intelligible fashion and written in standard English?

Reviewer #1: Yes

6. Review Comments to the Author

Reviewer #1: This revised version reads well. It is yet one more contribution to "equity" in health care literature

7. PLOS authors have the option to publish the peer review history of their article (what does this mean?). If published, this will include your full peer review and any attached files.

Reviewer #1: **Yes: **Johanne Sundby

---

## [Editor Report · Acceptance letter]

7 Apr 2022

PONE-D-21-15143R1 

Country of birth as a potential determinant of inadequate antenatal care use among women giving birth in Brussels. A cross-sectional study. 

Dear Dr. Schönborn:

I'm pleased to inform you that your manuscript has been deemed suitable for publication in PLOS ONE. Congratulations! Your manuscript is now with our production department. 

Kind regards, 

on behalf of

Dr. Angela Lupattelli 

Academic Editor

PLOS ONE